# Trends in *Pseudomonas aeruginosa* (*P. aeruginosa*) Bacteremia during the COVID-19 Pandemic: A Systematic Review

**DOI:** 10.3390/antibiotics12020409

**Published:** 2023-02-18

**Authors:** Qin Xiang Ng, Natasha Yixuan Ong, Dawn Yi Xin Lee, Chun En Yau, Yu Liang Lim, Andrea Lay Hoon Kwa, Ban Hock Tan

**Affiliations:** 1Health Services Research Unit, Singapore General Hospital, Singapore 169608, Singapore; 2MOH Holdings Pte Ltd., 1 Maritime Square, Singapore 099253, Singapore; 3Yong Loo Lin School of Medicine, National University of Singapore, 10 Medical Dr, Singapore 117597, Singapore; 4School of Medicine, Dentistry and Nursing, University of Glasgow, Glasgow G12 8QQ, UK; 5Department of Pharmacy, Singapore General Hospital, Singapore 169608, Singapore; 6Programme in Emerging Infectious Diseases, Duke-NUS Medical School, Singapore 169857, Singapore; 7Department of Infectious Diseases, Singapore General Hospital, Singapore 169608, Singapore

**Keywords:** *P. aeruginosa*, COVID-19, collateral effect, antibiotics, antimicrobial stewardship

## Abstract

*Pseudomonas aeruginosa* (*P. aeruginosa*) is among the most common pathogens associated with healthcare-acquired infections, and is often antibiotic resistant, causing significant morbidity and mortality in cases of *P. aeruginosa* bacteremia. It remains unclear how the incidence of *P. aeruginosa* bacteremia changed during the Coronavirus Disease 2019 (COVID-19) pandemic, with studies showing almost contradictory conclusions despite enhanced infection control practices during the pandemic. This systematic review sought to examine published reports with incidence rates for *P. aeruginosa* bacteremia during (defined as from March 2020 onwards) and prior to the COVID-19 pandemic. A systematic literature search was conducted in accordance with PRISMA guidelines and performed in Cochrane, Embase, and Medline with combinations of the key words (pseudomonas aeruginosa OR PAE) AND (incidence OR surveillance), from database inception until 1 December 2022. Based on the pre-defined inclusion criteria, a total of eight studies were eligible for review. Prior to the pandemic, the prevalence of *P. aeruginosa* was on an uptrend. Several international reports found a slight increase in the incidence of *P. aeruginosa* bacteremia during the COVID-19 pandemic. These findings collectively highlight the continued importance of good infection prevention and control and antimicrobial stewardship during both pandemic and non-pandemic periods. It is important to implement effective infection prevention and control measures, including ensuring hand hygiene, stepping up environmental cleaning and disinfection efforts, and developing timely guidelines on the appropriate prescription of antibiotics.

## 1. Introduction

*Pseudomonas aeruginosa* (*P. aeruginosa*), a Gram-negative bacterium, is a common opportunistic pathogen associated with healthcare acquired infections [1] and immunocompromised individuals. Importantly, it is often antibiotic-resistant and is a major cause of morbidity and mortality among hospitalized patients [2]. *P. aeruginosa* infection substantially increases overall healthcare costs and can lead to severe, life-threatening infection, especially in immunocompromised hosts [3,4]. The treatment of *P. aeruginosa* bacteremia typically involves the use of antibiotics, but in some cases, the infection may be difficult to eradicate and may require prolonged or intensive therapy. Despite advances in medicine and antibiotic therapy, *P. aeruginosa* infection still results in high mortality rates of up to 62% in certain patient groups [5]. The bacteria enter the bloodstream and can spread to various organ-systems, leading to serious and potentially life-threatening complications such as sepsis, organ failure, and shock.

Considerable attention was paid to the surveillance and detection of *P. aeruginosa* because it is a nosocomial pathogen that is highly adaptable and has evolved resistance to multiple antibiotics, is ubiquitous in water in sinks (and can contaminate breathing equipment), disproportionately affects immunocompromised hosts, and is the most serious bacteria causing ventilator-associated pneumonia [6,7,8]. Healthcare-associated infections caused by *P. aeruginosa* are also becoming more common, presenting as pneumonia, urinary tract infections, surgical site infections, and bacteremia, and with a prevalence of around 7% among all nosocomial infections [9,10]. This number is even higher in intensive care unit (ICU) settings as *P. aeruginosa* is an opportunistic pathogen with a predilection for immunocompromised patients [8].

As there are few options for empirical treatment in the ICU, this makes antibiotic resistance in *P. aeruginosa* an issue of serious concern. A common class of drugs in empirical and definitive treatment is carbapenems (such as imipenem, meropenem and, more recently, doripenem). However, these have been rendered ineffective in the face of carbapenem-resistant *Acinetobacter baumannii* (CRA) and *Enterobacteriaceae* (CRE), which have seen more frequent outbreaks in some places during COVID-19 [11]. This could be in part due to overstretched human and laboratory resources for COVID-19 diagnosis, treatment, and care, which reduces the capacity to screen for multidrug-resistant organisms (MDROs), and may produce lapses in traditional infection prevention and control practices and result in the inability to isolate or cohort all MDRO-positive patients [12]. These factors may result in an increase in antibiotic resistance, and to combat the rise of resistance, antibiotic stewardship is essential in not only the prescription of antibiotics but also their de-escalation [13], along with the surveillance efforts required to inform recommendations.

Globally, the rates of *P. aeruginosa* infections have been on a general upward trend [14,15]. This may be partially attributed to the increasing prevalence of risk factors for *P. aeruginosa* infections, such as an aging population, increase in chronic disease burden, increased use of medical devices, and an increase in the number of immunocompromised individuals. Infrequently, there were also reports of *P. aeruginosa* outbreaks, due to inadvertent lapses in infection control measures such as unclean or faulty medical equipment, or environmental reservoirs that went by undetected [16]; the bacterium is found in a variety of environments, including soil, water, and clinical specimens. Moist environments are associated with outbreaks of *P. aeruginosa*, and it is also known to be a difficult-to-treat multidrug-resistant organism [6,7,8], and the increasing prevalence of resistance to multiple classes of antibiotics may contribute to the incidence of *P. aeruginosa* bacteremia.

During the Coronavirus Disease 2019 (COVID-19) pandemic, improved hand hygiene and enhanced infection prevention and control measures were thought to positively influence the rates of several nosocomial infections in healthcare settings [17,18], but this finding was not universal. It remains unclear if the incidence of *P. aeruginosa* bacteremia increased, decreased, or remained stable during the pandemic, with studies showing contradictory conclusions [19,20,21]. Questions also remain as to whether the increased prescription of antibiotics during the pandemic [22] contributed to the emergence of resistant strains, especially since exposure to antibiotics is a primary risk factor leading to resistance, and data have found that resistant *P. aeruginosa* strains may emerge as early as eight days after the initiation of meropenem [23].

With this background in mind, we hypothesized that compared with the pre-pandemic period, the incidence of *P. aeruginosa* should have decreased, with stable antibiotic susceptibility patterns during the COVID-19 pandemic. As a significant amount of time has passed since the pandemic first began in early 2020, it is, therefore, opportune to reflect on the collateral effects of the pandemic, and this review sought to examine published reports with incidence rates for *P. aeruginosa* bacteremia during and prior to the COVID-19 pandemic.

## 2. Methods

The review protocol was prospectively registered in PROSPERO (registration number CRD42023387066). A systematic literature search was conducted in accordance with PRISMA guidelines [24] and performed in Cochrane, Embase, and Medline, using combinations of the search terms (pseudomonas aeruginosa OR PAE) AND (incidence OR surveillance), from database inception until 1 December 2022. The full search strategy for the various databases is shown in Table 1.

The inclusion criteria for the present review include: (1) original studies published in English, and (2) with reported incidence of *P. aeruginosa* bacteremia (based on a positive blood culture result) during and prior to the COVID-19 pandemic. This was defined to be from March 2020 onwards, the point when the World Health Organization (WHO) declared COVID-19 a global pandemic. Full texts were obtained for all articles of interest and their reference lists were hand-searched to identify additional relevant papers. Conflicts were resolved by discussion and consensus amongst four study investigators (Q.X.N., N.Y.O., C.E.Y., and Y.L.L.). 

A standardized data extraction form in Microsoft Excel was used to extract the relevant information from the studies reviewed. This was performed by three study investigators (Y.L.L., N.Y.O., and C.E.Y.) and cross-checked by a fourth (Q.X.N.) for accuracy.

As the number of available studies were limited and had dissimilar designs and diverse sources of data, this precluded the possibility of performing a meta-analysis.

As this study was a systematic review of published data, it did not require prior ethical approval.

## 3. Results

Of the 4604 initial search results, 1634 duplicates were removed. Then, 2914 studies were excluded at the title-abstract screening, and 48 more at the full-text screening due to the lack of reporting pre-post *P. aeruginosa* bacteremia data in the stipulated time period (Figure 1). Eight studies [19,20,21,25,26,27,28,29] were included in the final review, and their details and salient findings pertaining to our research question are summarized in Table 2. In the studies, the diagnosis of *P. aeruginosa* bacteremia was based on positive blood culture results.

There were two reports from Italy [26,27] and Turkey [21,29], and one report each from England [28], France [19], Japan [20], and Serbia [25]. None of the reports contained data from the year 2022 onwards. Four of the reports found higher incidence of *P. aeruginosa* bacteremia during the pandemic period (as compared to non-pandemic periods) [19,26,27,28], with two reporting no change in trend [21,25] and two reporting decreases in the incidence of *P. aeruginosa* bacteremia [20,29].

It was noteworthy that antibiotic susceptibility patterns for *P. aeruginosa* appeared to remain stable [19,26,28], although one report found a higher incidence of ceftazidime-resistant strains [19] with several reports documenting increased antibiotic prescription during the COVID-19 pandemic period [19,27]. Nonetheless, antibiotic resistance patterns are known to change over time and vary significantly based on geography and the type of infection [30]. 

## 4. Discussion

Prior to the COVID-19 pandemic, the prevalence of *P. aeruginosa* was on an uptrend [15,16]. However, the incidence rates for *P. aeruginosa* bacteremia during the COVID-19 pandemic remained contested, with four reports showing a higher incidence of *P. aeruginosa* bacteremia during the pandemic period (as compared to non-pandemic periods) [19,26,27,28], two reporting no change in trend [21,25], and two reporting decreases in the incidence of *P. aeruginosa* bacteremia [20,29]. 

A meta-analysis of 144 published studies from 2005 to 2016 found that, irrespective of a country’s income level, a significant proportion of healthcare-associated infections (35 to 55%) were in fact preventable [31]. This implies that there is both great potential to further reduce the burden of nosocomial infections and existing gaps in the implementation of infection control practices. During the COVID-19 pandemic, there was an increased demand for environmental services workers [32], and infection control measures in hospitals were thought to be enhanced during the COVID-19 pandemic, which should have theoretically reduced the incidence of hospital-acquired infections such as *P. aeruginosa* and *Clostridioides difficile* infections [33]. İpek et al. noted a decline in the incidence of *K. pneumonia* in their pediatric ICU, and did not see cases of *P. aeruginosa* or *Enterococcus fecium*. They attributed this remarkable finding to a rise in the hand hygiene rate, which was above 99% during the pandemic, while it averaged 94% before the pandemic [21]. *P. aeruginosa* bacteremia can be prevented through good hygiene practices. Improvements in infection control measures included hand hygiene, appropriate use of personal protection equipment (PPE), and an increased focus on environmental decontamination, all of which aimed to reduce the possibility of contact transmission and other nosocomial spread. However, it is also possible that prioritizing respiratory infections may have had unintended effects, as per the experiences of other centers [34], and other infection control measures may have been compromised during the pandemic.

There are recent reports of decreased compliance to hand hygiene [34] and significantly increased central-line-associated bloodstream infections (CLABSI) [35] during the COVID-19 pandemic. Although good hand hygiene, i.e., the practice of consistently and effectively washing or sanitizing one’s hands, is thought to be a cornerstone of infection control and prevention, data collected using an electronic hand hygiene monitoring system in two Danish hospitals found hand hygiene compliance was lower during the COVID-19 pandemic as compared to pre-pandemic periods [36,37]. In one of the studies, despite an initial improvement in hand hygiene compliance, healthcare workers appeared to revert to old routines once data presentation meetings on hand hygiene rates and hand hygiene related initiatives were stopped [36]. It is evident that hand hygiene compliance is not a one-time event, but requires a continuous process of ongoing education, monitoring, and improvement within the institutions. In fast-paced healthcare environments, healthcare providers may not have the time to wash their hands as often as recommended, which can impact compliance rates. Further to hand hygiene, data from 148 US hospitals also showed a rise in CLABSI, catheter-associated urinary tract infection, and MRSA bacteremia [38]. The authors attributed the rise to the additional burden of COVID-19 care disrupting routine practice, and pointed to lapses in usual infection prevention practices [38]. Lapses in optimal line care and infection control could have been aggravated by poorer staffing and an increased patient load during the pandemic [39]. For example, the intensive care unit capacity in some hospitals was expanded without an accompanying rise in skilled staff and equipment [40]. High patient volume, even without exceeding capacity, affects patient care [39], and this could have compromised certain infection control practices in the hospital. In mid-2020, there was also the problem of shortage of PPE in some parts of the world. A lack of resources, including funding and personnel, can impact the implementation and maintenance of effective hand hygiene programs. This may also have led to unintentional circumventions and contributed to the spread of nosocomial pathogens [41]. Self-contamination was also a frequently encountered problem associated with incorrect doffing procedures of PPE worn by healthcare providers during the pandemic [42]. In the case of *P. aeruginosa*, the bacterium can be easily spread from person to person or from contaminated medical equipment and surfaces, and can persist in the environment for long periods of time. *P. aeruginosa* bacteremia is a serious and potentially life-threatening condition that is associated with significant morbidity and mortality [2,3,4]. Regular monitoring and feedback, as well as ongoing education and improvement efforts, can help to close any gaps identified and improve infection control practice over time.

Separately, patients with moderate or severe COVID-19 illness appear predisposed to increased risk of hospital-acquired bloodstream infections [43,44]. A few reports suggested that COVID-19 patients seemed to be more susceptible to co- or secondary infections, [28], caused by Gram-negative bacteria such as *P. aeruginosa*, which was the second most common pathogen. This could be related to the use of steroids and other immunomodulators in critically ill COVID-19 patients. Other contributing factors included the longer hospitalization of severely ill COVID-19 patients and a higher risk of receiving invasive devices or admission to intensive care units [45]. Furthermore, for critically ill patients who require mechanical ventilation, *P. aeruginosa* is also the most common multidrug-resistant Gram-negative pathogen [46]. The use of invasive medical devices, such as ventilators and catheters, can also increase the risk of *P. aeruginosa* infections in hospital settings.

The epidemiology of *P. aeruginosa* infections varies depending on the population and healthcare setting, but several factors have been associated with an increased risk of infection. Importantly, the rise in the prescription of antibiotics in some countries during the pandemic [19,27,47] could have contributed to the observed trends. In particular, COVID-19 patients had high rates of antibiotic prescription and tended to receive antibiotics in view of chest radiograph changes since it is difficult to be certain that there is no concomitant or secondary bacterial infection [48,49]. Moreover, in the early stages of the COVID-19 pandemic, there was much uncertainty about the epidemiology and characteristics of the SARS-CoV-2 virus, coupled with a rapid increase in case numbers, a lack of clear treatment protocols, and the suspicion of nosocomial infections in patients with prolonged hospitalization [47]. Furthermore, there was the added complexity arising from the similarity between COVID-19 and pneumonia, in terms of clinical presentation and radiological and laboratory test results [50]. Additionally, drawing from past knowledge of respiratory viruses, viral outbreaks such as influenza were correlated with a rise in co-infections by bacterial pathogens; a meta-analysis by Klein et al. found that most studies fell within the range of 11 to 35% [51]. As a result, it was perhaps unsurprising that a retrospective analysis of 17 hospitals in South Carolina, United States, found a significant increase in overall and broad-spectrum antibiotic use in seven hospitals admitting patients with COVID-19 [22]. Furthermore, a 2022 meta-analysis of 19 studies found an overall high antibiotics consumption of 68% among COVID-19 patients [48], and this was particularly an issue among lower and middle-income countries. Further research has demonstrated that only a small minority of hospitalized COVID-19 patients suffered from bacterial or fungal co-infections [49,52], with 7% of patients having superimposed bacterial infection [49], less than other respiratory infections such as influenza. At the same time, in the community, antibiotics prescription as empiric treatment has also been deemed excessive, particularly in general practice [53,54]. This inappropriate use of antibiotics is concerning, as it exerts a selection pressure on the bacteria, selecting for strains that have developed resistance. Since the indiscriminate use of antibiotics contributes to resistance over time, this demands the development of timely clinical practice guidelines and appropriate antimicrobial stewardship interventions, even during a pandemic period. *P. aeruginosa* has high intrinsic resistance to antibiotics as well as a remarkable capacity to acquire new resistance mechanisms [55]. Of concern is the observed increased use of antibiotics without consultation and culture testing during the COVID-19 pandemic. Fortunately limited to a single report [56], this is a practice that must not be allowed to propagate. For septic patients who require antibiotic therapy, it is vital to first obtain blood cultures before antibiotic administration, and continually review the indication and use of antibiotics and stop or de-escalate antibiotics when appropriate to do so. Over the years, multifaceted interventions including improvements in antibiotic stewardship and surveillance have achieved some success in reducing the rates of nosocomial infections [31] and countering the problem of antibiotic resistance [57], and these efforts should be sustained.

As a proposal for effective institutional antimicrobial stewardship in a pandemic situation, a multi-disciplinary team effort involving doctors, pharmacists, nurses, and patient educators can be adopted. On a wider scale, recommendations should be directed by good collaboration across medical specialties such as public health, preventative medicine, infectious diseases, and microbiology.

Another possible explanation for the observed trend of higher incidence of *P. aeruginosa* bacteremia during the pandemic period is the longer length of stay of severe or moderately severe COVID-19 cases [44], especially for patients who acquire *P. aeruginosa* nosocomial infections. An increased length of stay was highly associated with the risk of acquisition of nosocomial infections [58], which in turn also resulted in poorer outcomes.

Nevertheless, limitations of the present review include the small number of reports and heterogeneous nature of the studies, which precluded the feasibility of performing a meta-analysis. Second, the incidence of *P. aeruginosa* bacteremia is subject to some temporal variations [59,60], and the studies covered relatively short periods, while the pandemic has lasted three years (and is still not officially over). The circumstances and context changed as the pandemic progressed, and a time-sequence analysis might provide useful statistical information and characteristics and be more enlightening. Third, the collateral effects of the pandemic may also not be fully realized until further longitudinal studies become available. The reports in the present review only contained data from 2019 to 2021. There is a need for continual, close monitoring. Fourth, the extent of surveillance efforts may have been limited during the pandemic, partly due to lower testing rates, manpower shortages, fears of disease transmission, etc. [61]. This may affect the data collected and our interpretations. Finally, it is also worth studying the outcomes of patients with *P. aeruginosa* bacteremia as early reports have hinted at the possibility of delayed diagnosis and delayed treatment [61] due to higher patient load and higher stress levels experienced by the medical staff during the COVID-19 pandemic. This is of clinical significance and should be the subject of future research, especially since the early recognition and administration of appropriate treatment of *P. aeruginosa* bacteremia is associated with better outcomes and lower rates of morbidity and mortality [62].

## 5. Conclusions

In conclusion, several international reports found a slight increase in the incidence of *P. aeruginosa* bacteremia during the COVID-19 pandemic. These findings run counter to our initial hypothesis, and they emphasize the continued importance of good infection prevention and control and antimicrobial stewardship during pandemic and non-pandemic periods. To reduce the risk of *P. aeruginosa* bacteremia and other infections, it is important to implement effective infection prevention and control measures, including ensuring hand hygiene, stepping up environmental cleaning and disinfection efforts, and developing timely guidelines on the appropriate prescription of antibiotics. These are important lessons for future pandemic planning. As part of ongoing antimicrobial stewardship and surveillance efforts, these trends should be further monitored and studied. Questions also remain as to whether these patients suffered worse outcomes during COVID-19 due to certain collateral effects of the pandemic, and this should be the focus of future investigations.

## Figures and Tables

**Figure 1 antibiotics-12-00409-f001:**
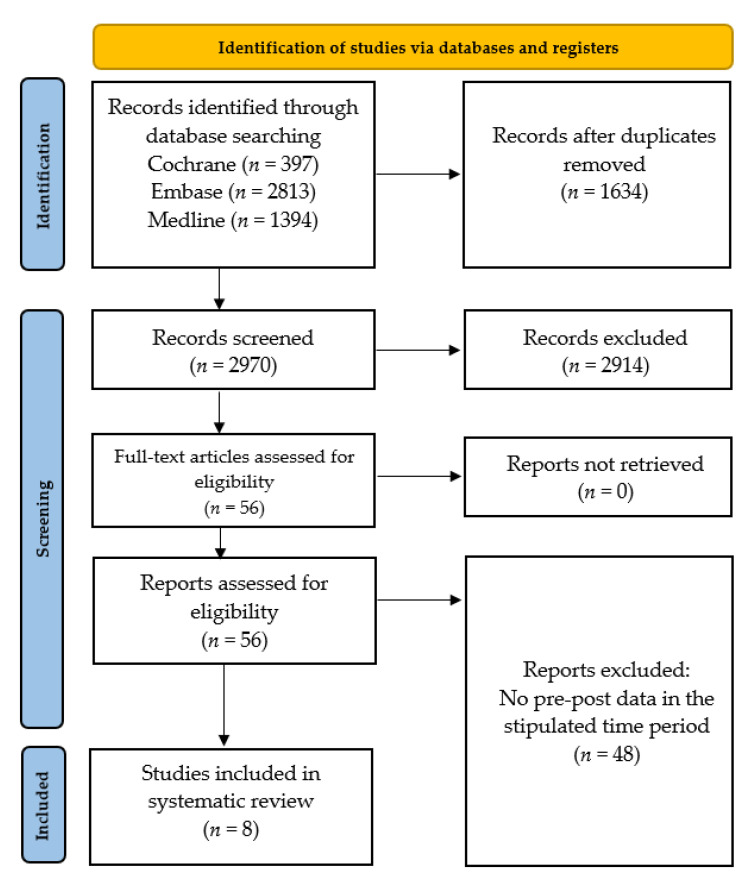
PRISMA flowchart showing the study abstraction process.

**Table 1 antibiotics-12-00409-t001:** Full search strategy for Medline, Embase, and Cochrane databases.

**Medline**	
1	(Incidence or surveillance).ti,ab. or exp “incidence”/
2	(pseudomonas aeruginosa or PAE).ti,ab. or exp “pseudomonas aeruginosa”/
3	1 and 2
4	limit 3 to (english language and year = “2019 to 2023”)
**EMBASE**	
1	Incidence:ti,ab OR surveillance:ti,ab OR ‘incidence’/exp
2	‘pseudomonas aeruginosis’:ti,ab OR ‘pseudomonas aeruginosa’/exp OR ‘pae’:ti,ab
3	#1 AND #2
4	#1 AND #2 AND [English]/lim AND [2019 to 2023]/py
**Cochrane Database**	
#1	(incidence or surveillance):ti,ab,kw AND (“pseudomonas aeruginosa” or PAE):ti,ab,kw
#2	Incidence or surveillance
#3	(incidence or surveillance):ti,ab,kw
#4	MeSH descriptor: [Incidence] explode all trees
#5	MeSH descriptor: [Pseudomonas aeruginosa] explode all trees
#6	(pseudomonas aeruginosa OR PAE):ti,ab,kw
#7	#5 OR #6
#8	#5 AND #6
#9	#3 OR #4
#10	#7 AND #9

**Table 2 antibiotics-12-00409-t002:** Characteristics of the studies reviewed (arranged alphabetically by the first author’s last name).

Study (Year)	Country	Setting	Time Periods Compared	Key Findings
Amarsy, 2022 [19]	France	Multihospital institution	Jan–Apr 2019 and Jan–Apr 2020	- higher incidence of bloodstream infection, ceftazidime-resistant strains of *P. aeruginosa* (2.4-fold increase) - increased antibiotic prescription during the pandemic period
Despotovic, 2022 [25]	Serbia	Adult ICU, single-center	Apr 2019–Apr 2021	- a total of three *P. aeruginosa* bacteremia cases were recorded from 2019 to 2021, with no significant change in trend (*p* = 0.23)
Hirabayashi, 2022 [20]	Japan	1300 hospitals with ≥200 beds	Jan–Sep 2019 and Jan–Sep 2020	- decrease in incidence of *P. aeruginosa* by 7.2% between second quarter of 2019 and 2020, and by 3.6% between the third quarter - decrease in incidence of carbapenem-resistant *P. aeruginosa* as well
İpek, 2022 [21]	Turkey	Paediatric ICU, single-center	Apr–Sep 2019 and Apr–Sep 2020	- there were 5 cases of *P. aeruginosa* observed during the pre-pandemic period and 0 during the pandemic period
Meschiari, 2022 [26]	Italy	University hospital, single-center	Jan 2015–Feb 2020 and Mar 2020–Nov 2021	- decrease in the trend of all antibiotic use during pandemic period - increase in bloodstream infection due to carbapenem-susceptible *P. aeruginosa* (*p* = 0.032) but not carbapenem-resistant strains (*p* = 0.406)
Shbaklo, 2022 [27]	Italy	Tertiary hospital, single-center	Aug 2019–Feb 2020 and Feb 2020–Mar 2021	- slight increase in the incidence rate ratio of *P. aeruginosa* bacteremia compared to pre-pandemic period (0.06 vs. 0.09, *p* = 0.96) - increased use of antibiotics (fourth- and fifth-generation cephalosporins and piperacillin-tazobactam) in the first wave of the COVID-19 pandemic
Sloot, 2022 [28]	England	NHS acute trusts	Aug 2020 and Feb 2021	- increase in incidence from 4.9 (N = 139, 95% CI 4.1 to 5.7) per 100,000 bed-days in Aug 2020 to 6.2 (N = 164, 95% CI 5.3 to 7.2) per 100,000 beddays in Feb 2021, coinciding with co- or secondary infections to COVID-19 cases - increases were seen for *P. aeruginosa*, *Klebsiella* spp. bacteremia but not for *E. coli* bacteremia - little variation in terms of antibiotic susceptibility results
Yardimci, 2022 [29]	Turkey	Tertiary hospital, single-center	Jan 2016–Dec 2020	- increase in incidence from 2016 to 2019 but decreased during the COVID-19 pandemic - 13 cases (6.5%) in 2016, 17 cases (6.6%) in 2017, 36 cases (12.3%) in 2018, 37 cases (11.8%) in 2019 and 22 cases (10.5%) in 2020

Abbreviations: COVID-19, Coronavirus Disease 2019; ICU, intensive care unit; NHS, National Health Service.

## Data Availability

The datasets generated during and/or analyzed during the current study are available from the corresponding author on reasonable request.

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
