# Peer review of "Trends in Pseudomonas aeruginosa (P. aeruginosa) Bacteremia during the COVID-19 Pandemic: A Systematic Review"

_antibiotics, 2023, doi:10.3390/antibiotics12020409_

Round 1

Reviewer 1 Report

Antibiotic-resistant infections, especially in the hospital environment, are extremely dangerous. Assessing the impact on the quality of care and life of patients, especially during a pandemic, is extremely important. P. aeruginosa, due to the decrease in immunity during COVID-19, as an antibiotic-resistant bacterium, is extremely dangerous. From this point of view, the average review is undoubtedly relevant.

In my opinion, the work is missing:

1. Fundamental mathematical statistics, and such parameters as the student's coefficient, groupings of averages.

2. Graphic material such as: cluster, discriminant, factor and component analysis.

3. And of course the Monte Carlo method, as the best way to describe random processes. Since it is impossible to predict which of the patients will get sick.

Author Response

Thank you for the comments. We have acknowledged in our discussion of study limitations that, "Second, the incidence of P. aeruginosa bacteremia is subject to some temporal variations [34,35], and the studies covered relatively short periods, while the pandemic lasted three years (and is still not officially over). Circumstances changed as the pandemic progressed, and a time-sequence analysis might provide useful statistical information and characteristics and be more enlightening."

Reviewer 2 Report

Dear editorial office

I have read and reviewed the paper by Xiang Yu et al entitled “Trends in Pseudomonas aeruginosa (P. aeruginosa) Bacteremia During the COVID-19 Pandemic: A Systematic Review” submitted to the antibiotics.

The reported that prior to the COVID-19 pandemic, the prevalence of P. aeruginosa was on an uptrend. However, different factors related to this pandemic has changed the profile of our overview regarding this bacterial infection.

I think that paper is well written and it can be considered in this journal after a minor correction.

1-      title is fine and enough informative, I feel.

2-      Abstract is too long so I recommend to shortening it.

3-      Introduction is concise and short enough.

4-      Methods: who confirmed the bacteremia to be included in P. aeruginosa infected subjects? This is important! An infectious expert? Or? You have huge number of papers that needs to be examined with same criteria. Please indicate your criteria’s that involved in this section. I think you had some people involved clinical settings.

5-      Change the “Of the 4604 initial” to “Of 4604 initial ” in results section.

6-      The trend of data gathering in PRISMA flowchart is fine and acceptable.

7-      The clinical relevance of sentence “It was noteworthy that antibiotic susceptibility patterns for P. aeruginosa appeared to remain stable” stated in the results section should be clearly explained in your discussion. Why? Why not changed? At least it should be changed since the number of studies occurred in this duration was drastically changed! It can be happened but statistically it seems impossible. At least to my opinion. Please clarify it more.

8-      In the results section, I saw that a lot of references are exist related to the compared researches. I should assert that here is not necessary to compare anything together and remove them all and just go for them in discussion section. In the results section, just pay attention to the your data.

Author Response

  1. We have made the text edits as suggested by the reviewer.
  2. We have now shortened the abstract to 198 words as advised by the reviewer.
  3. Thank you for the comments. We have now added that, "In the studies, the diagnosis of P. aeruginosa bacteremia was based on positive blood culture results."
  4. Thank you for the comment. We apologise for the ambiguous word choice and have now explained that "It was noteworthy that antibiotic susceptibility patterns for P. aeruginosa appeared to remain stable [9,14,16], with only one report finding a higher incidence of ceftazidime-resistant strain".
  5. Thank you for the comment. We have paid closer attention to our data as advised by the reviewer.

Round 2

Reviewer 1 Report

The authors fully answered the questions I asked.